# Staging Computed Tomography Parameters Predict the Need for Vein Resection during Pancreaticoduodenectomy in Resectable Pancreatic Ductal Adenocarcinoma

**DOI:** 10.3390/diagnostics14020135

**Published:** 2024-01-07

**Authors:** Rupaly Pande, Wingyan Liu, Syed S. Raza, Michail Papamichail, Arul E. Suthananthan, David C. Bartlett, Ravi Marudanayagam, Bobby V. M. Dasari, Robert P. Sutcliffe, Keith J. Roberts, Sharan Wadhwani, Nikolaos Chatzizacharias

**Affiliations:** 1Department of HPB and Liver Transplant Surgery, Queen Elizabeth Hospital, University Hospitals Birmingham NHS Foundation Trust, Birmingham B15 2TT, UK; rupaly.pande@uhb.nhs.uk (R.P.); syed.raza@uhb.nhs.uk (S.S.R.); michail.papamichail@uhb.nhs.uk (M.P.); aruledward@gmail.com (A.E.S.); david.bartlett@uhb.nhs.uk (D.C.B.); ravi.marudanayagam@uhb.nhs.uk (R.M.); bobby.dasari@uhb.nhs.uk (B.V.M.D.); robert.sutcliffe@uhb.nhs.uk (R.P.S.); keith.roberts@uhb.nhs.uk (K.J.R.); 2Department of Radiology, Queen Elizabeth Hospital, University Hospitals Birmingham NHS Foundation Trust, Birmingham B15 2TT, UK; winngyan.liu@nhs.net (W.L.); sharan.wadhwani@uhb.nhs.uk (S.W.); 3Institute of Immunology and Immunotherapy, University of Birmingham, Birmingham B15 2TT, UK; 4College of Medical and Dental Sciences, University of Birmingham, Birmingham B15 2TT, UK

**Keywords:** PDAC, pancreaticoduodenctomy, tumour–vein interface, vein resection

## Abstract

Background: Surgery-first approach is the current standard of care for resectable pancreatic ductal adenocarcinoma (PDAC), and a proportion of these cases will require venous resection. This study aimed to identify parameters on staging computed tomography (CT) that predict the need for venous resection during pancreaticoduodenectomy (PD) for resectable PDAC. Methods: We conducted a retrospective analysis of prospectively collected data on patients who underwent PD for resectable staged PDAC (as per NCCN criteria) between 2011 and 2020. Staging CTs were independently reviewed by two specialist radiologists blinded to the clinical outcomes. Univariate and multivariate risk analyses were performed. Results: In total, 296 PDs were included. Venous resection was performed in 62 (21%) cases. There was a higher rate of resection margin positivity in the vein resection group (72.6% vs. 48.7%, *p* = 0.001). Tumour at the neck of the pancreas, superior mesenteric vein involvement of ≥10 mm and pancreatic duct dilatation were identified as independent predictors for venous resection. Discussion: Staging CT parameters can predict the need for venous resection during PD for resectable cases of PDAC. This may assist in surgical planning, patient selection and counselling. Future efforts should concentrate on validating these results or identifying additional predictors in a multicentre and prospective setting.

## 1. Introduction

Pancreatic ductal adenocarcinoma (PDAC) is an aggressive cancer with survival as low as 7% at 5 years [1]. Initial staging, with three-dimensional imaging, is crucial to determine the appropriate management pathway for each stage of the disease. Use of computed tomography (CT) as the primary mode of imaging is recommended by the National Comprehensive Cancer Network (NCCN) and European Society for Medical Oncology (ESMO) [2,3]. Clinical guidelines utilise the extent of vascular involvement on CT to stage the disease as either resectable, borderline resectable or locally advanced, with suggested treatment pathways for each stage. Regarding venous involvement, staging is based on the circumferential extent of tumour abutment and any changes in vein contour. Tumours are staged as resectable if there is no abutment of the portal (PV) and superior mesenteric (SMV) veins or if abutment is ≤ 180 degrees with no change in contour or vein wall irregularity; additionally, no arterial contact should be present. Tumours are staged as borderline resectable if there is radiological evidence of abutment > 180 degrees of the veins, contour irregularity or thrombosis, provided venous resection and reconstruction is technically possible. Otherwise, tumours are upstaged to locally advanced. Borderline resectable and locally advanced cases are high risk for positive resection margins and neoadjuvant therapy followed by resection is the suggested treatment [2,3]. For resectable cases, the surgery-first approach is the standard of care. Nonetheless, the CT criteria for the extent of vascular involvement, which affect staging and hence treatment pathways, remain controversial, lacking in evidence and open to interpretation and interobserver variability by radiologists and surgeons.

With advances in peri-operative care and surgical technique, the criteria of operability have changed over the years. Currently, venous involvement is not considered a contraindication to resection, but rather a standard component of pancreatic resectional surgery. Peri-operative outcomes are comparable to cases where venous resection is not required [4,5]. A recent meta-analysis [6] revealed that patients who underwent venous resection exhibited worse pathological features, such as higher rates of resection margin positivity, lymph node involvement and a larger tumour size. However, overall survival was not found to be significantly different, and 1-, 3- and 5-year survival rates were better in those who underwent venous resection. Short-term outcomes were also comparable (with the exception of haemorrhage) in terms of overall complication rates, pancreatic fistula, delayed gastric emptying, need for reoperation and 90-day mortality. Pre-operative surgical planning should always account for vein resection in borderline and locally advanced cases, and vein resection may often be required in cases initially staged as resectable based on imaging. Therefore, during staging investigations, identifying any pre-operative factors on imaging that could predict the need for venous resection in these cases would be of significant benefit for surgical planning, as well as appropriate patient selection and counselling.

This study aims to identify radiological parameters on pre-operative CT staging scans that can predict the need for venous resection during pancreaticoduodenectomy (PD) for resectable cases of PDAC.

## 2. Materials and Methods

This was a retrospective cohort study conducted in line with STROBE (Strengthening the Reporting of Observational studies in Epidemiology) guidelines [7], following departmental approval at the University Hospitals of Birmingham, a tertiary specialist centre for the treatment of pancreatic cancer. All patients with PDAC who had PD (pylorus-preserving or classical) with or without venous resection were identified from a prospectively maintained database over a 10-year period (2011–2020). Patients with resectable PDAC as per NCCN criteria [3] were included in this study, specifically those with no contact with the vein or contact less than 180 degrees without change in vein wall contour or irregularity. Figure 1 presents examples of venous involvement (Figure 1c is staged as borderline resectable and was not included in this cohort). Patients with ampullary tumours, duodenal cancer, distal cholangiocarcinoma or borderline or locally advanced pancreatic tumours and patients who underwent total pancreatectomy were excluded. Pre-operative staging investigations included CT of the thorax, abdomen and pelvis and endoscopic ultrasound (EUS) with fine-needle aspiration (FNA) when pre-operative cytological diagnosis was required. Magnetic resonance imaging (MRI) and positron emission tomography (PET-CT) were used selectively if there were concerns of metastatic disease based on the CT scan. All cases were referred for adjuvant chemotherapy after surgical treatment and management of all cases was discussed and agreed in hepatopancreaticobiliary (HPB) multidisciplinary meetings. Demographic, clinical and pathological data were obtained from the hospital’s electronic records and the department’s prospectively maintained database. Vein resections were classified according to the system proposed by Tseng et al. [8]. Pre-operative CT images were independently reviewed by two HPB specialist radiologists (WL and SW), who were blinded to the operative, clinical and pathological outcomes. Any conflicts were resolved by further review and discussion with the rest of the authors.

The following radiological parameters were recorded: tumour location and size, presence of pancreatitis, common bile duct or pancreatic duct dilatation (≥4 mm defined as dilated), size of pancreatic duct, presence of regional lymphadenopathy, abutment of the PV, SMV or portomesenteric confluence, length of involvement of PV and SMV (in mm), circumference of abutment of PV and SMV (0–90 and 90–180 degrees) and time from CT to operation (in days).

Pathological analysis of specimens was performed in line with the Royal College of Pathologists dataset for histopathological reporting of cancers of the pancreas [9], and the TNM 8th edition [10] was used for staging. Margins were considered positive when malignant cells were identified within 1 mm from the resection margin in paraffin-embedded specimens.

### Statistics

The characteristics of the cohort were determined with standard descriptive statistical analysis. Chi-Square and Mann–Whitney U tests with exact statistics were used to compare nominal and ordinal variables with a significance level set at *p* < 0.05. For comparison of continuous variables, one-way analysis of the variance (ANOVA) was used with a significance level set at *p* < 0.05. Univariate and multivariate time-to-event analyses were performed using the Cox proportional hazard model to determine risk factors for venous resection with time defined as the period from CT to operation. Continuous parameters were also analysed as categorical. Variables were initially subject to a univariate analysis, and those with *p* < 0.2 were introduced into a multivariate model. Hazard ratios (HR) and associated 95% confidence intervals (CI) were calculated and a two-tailed *p* value < 0.05 was considered statistically significant. All statistical analyses were performed using the software package SPSS Statistics for Windows (version 25.0; SPSS Inc., Chicago, IL, USA).

## 3. Results

A total of 296 PD cases were included in this study, with venous resection performed in 62 (21%) cases. In the majority of those cases, a wedge resection with (*n* = 4, 7%) or without (*n* = 13, 21%) use of patch repair or primary end-to-end reconstruction (*n* = 36, 58%) was performed. An interposition graft was required in seven cases (11%) (in two cases, the type of venous resection was not recorded). The median age was higher in the vein resection group than those who did not undergo vein resection (69 (IQR 63–75) vs. 73 (IQR 67–77), *p* = 0.019), while more females had vein resection (59.7% vs. 40.3%, *p* = 0.045). There were no differences in the TNM staging and lymph node positivity between the groups, while a higher rate of resection margin positivity was identified in the vein resection group (48.7% vs. 72.6%, *p* = 0.001). Among the CT characteristics analysed, involvement of PV (*p* = 0.013), length (*p* = 0.012) and circumference (*p* = 0.019) were significantly higher in the vein resection group (Table 1).

The results of our univariate and multivariate time-to-event analyses of the CT parameters can be found in Table 2.

Tumour location at the neck of the pancreas, SMV involvement ≥ 90 degrees and the presence of a dilated pancreatic duct were independent predictors for venous resection. Though lymphadenopathy exhibited a trend, it did not reach statistical significance.

There were no parameters that could independently predict the need for a reconstruction with the use of an interposition graft despite our univariate analysis indicating this was more common in cases with CT evidence of pancreatitis (Table 3).

## 4. Discussion

Accurate staging is the foundation of appropriate management in surgical oncology. For non-metastatic pancreatic cancer, a consensus exists amongst clinical guidelines regarding the management of the different stages of the disease. Neoadjuvant treatment followed by resection in appropriate cases is advocated for borderline resectable and locally advanced stages, whilst a surgery-first approach is advocated for resectable cases. Guidance for defining resectability substantially varies not only by society but also within a society over time and between clinical trials and cohort studies. The NCCN resectability criteria, adapted from the Society of Abdominal Radiology and the American Pancreatic Association consensus statement, are recommended by the ESMO and NCCN guidelines [2,3]. According to the NCCN guidelines, R0 resection rates for resectable, borderline resectable and locally advanced PDAC are 73%, 55% and 16%, respectively [3].

Amongst the several classifications, resectability is determined by the extent of vascular involvement and the possibility of vascular reconstruction. With respect to venous involvement, the tumour–vein contact is assessed in terms of length and circumference, contour abnormalities and the presence of vascular infiltration or thrombosis. CT is the primary and most commonly used imaging technique to accurately determine staging and response to treatment. Although multi-detector CT has been shown to have 89–91% sensitivity and 85–90% specificity for pancreatic cancer, its accuracy in terms of correctly assessing tumour involvement of adjacent structures is limited [11,12,13,14]. Even in radiologically staged resectable PDAC, vascular involvement is an important determinant of irresectability at surgery in 50–65% of cases [15,16]. Within the literature, the accuracy of CT in terms of correctly predicting venous invasion in line with various classifications ranges from 62% to 92%, with 20% of cases deemed resectable according to the NCCN criteria exhibiting vein invasion on histology [17,18,19]. Furthermore, interobserver variations in the description and interpretation of scans may result in variability in staging and therefore management. Studies assessing interobserver agreement of resectability based on the NCCN criteria have demonstrated variation, which was particularly an issue in borderline resectable cases where difficulty in differentiating abutment from encasement or tumours from inflammation added to the interobserver variability [20,21,22]. A further study, which assessed the applicability of the NCCN guidelines in PDAC, showed that 10% of cases were not adequately described according to the guidelines, of which more than half were re-graded as borderline resectable due to additional tumour involvement [23]. As there is substantial variation in the definition for resectability, clear descriptions of the tumour–vein interface would better clarify and aid pre-operative planning [24].

Surgical resection is the only potentially curative management option for non-metastatic pancreatic cancer. For tumours of the head, uncinate process and proximal neck, PD, either classical or pylorus-preserving, is the procedure of choice. In contrast to past practices, venous resection is currently considered safe and a necessary requirement in the technical armoury of every pancreatic surgeon. However, variations in surgical experience and expertise may result in cases of venous involvement being deemed inoperable in less experienced hands. Although venous resection should be anticipated and part of pre-operative planning in borderline resectable and locally advanced tumours, in the majority of resectable cases, venous resection is not required. However, in our study, this was deemed necessary at intra-operative dissection in around 21% of cases. Amongst the pre-operative CT parameters, involvement of the portal vein by the tumour was more common in the vein resection subgroup. Intra-operatively, differentiating between PV involvement directly due to the tumour and that created by an inflammatory response around the tumour is difficult. Therefore, PV resection is justified if dissection of the vein is not possible or if the level of concern is high.

The majority of venous resection cases required a relatively simple wedge resection or full resection of the circumference of the vein with primary end-to-end reconstruction. The latter is only possible when adequate vein length is available, which has been reported to be between 20 mm and 40 mm [25,26]. Where inadequate length remains for reconstruction, an interposition graft is required; in our study, this was the case for 11% of the venous resection subgroup (2.6% of the complete cohort). Irrespective of the different types of grafts available, the need for an interposition graft increases the complexity of the operation. Therefore, identifying cases in a pre-operative setting where venous resection may be required, particularly where complex reconstruction may be necessary, may improve peri-operative planning or even result in referral of these cases to centres with higher volumes and more experience.

Risk model time-to-event analysis was performed to identify any parameters on CT that could predict the need for venous resection. This showed that neck tumours had more than three times the risk of requiring a venous resection compared to head or uncinate process tumours. Furthermore, tumour abutment of the SMV for 1 cm or more increased the risk by 3.5 times, while PD dilatation almost doubled the risk. A study by Phoa et al. demonstrated similar results, concluding that tumour–portal vein contact of more than 5 mm was an important determinant [27]. The same study also reported that circumferential involvement of the SMV/PV > 90 degrees was an important determinant, as opposed to in our study. Imamura et al. also differ in their cut-off value as they have demonstrated that PV/SMV contact > 20 mm and PV/SMV contact > 180 degrees are associated with higher vein resection rates and that PV/SMV contact > 20 mm is correlated with worse overall survival [28]. However, that study included borderline resectable cases and outcomes were not stratified by a neoadjuvant treatment regimen. A common theme across many studies is that the current determination of resectability based on the angle of PV/SMV is inadequate, and that a more reliable factor for vein resection rate included a greater length of vein contact and longer PV/SMV narrowing [29,30,31,32]. Whether any venous contact in resectable cases should upstage these to borderline is under debate, as SMV/PV contact in resectable cases has been identified as an independent predictor of overall and disease-free survival [33].

With regards to cases that required complex reconstruction with the use of an interposition graft, the only pre-operative CT parameter that trended towards but did not achieve significance was the presence of pancreatitis. This is not surprising as pancreatitis substantially increases the complexity of PD due to the loss of planes from concomitant peripancreatic inflammation [34,35], and it is likely that with larger numbers, this would have reached significance. Therefore, the current preferred approach in our unit, as well as others as published in the literature [35], is to treat patients with pancreatic cancer and pancreatitis with neoadjuvant chemotherapy even in the presence of a resectable stage of disease.

Limitations of our study include its retrospective single-centre nature, even though the data were collected prospectively. An element of time bias may also have been present due to the decade-long time span of the study and the significant improvements in CT imaging and available software tools to examine the tumour–venous interface, such as high-resolution, thin-slice and multiplanar reconstruction techniques. Nonetheless, this is one of the largest cohort studies that has been reported on the subject, from a tertiary high-volume unit with uniform practice utilising a surgery-first approach for resectable pancreatic cancer cases, over the years.

## 5. Conclusions

In conclusion, the prognostic factors identified in this study may aid pre-operative planning by predicting the need for venous resection during PD in resectable cases of pancreatic cancer. Future efforts should concentrate on validating these results or identifying additional predictors in a multicentre and prospective setting.

## Figures and Tables

**Figure 1 diagnostics-14-00135-f001:**
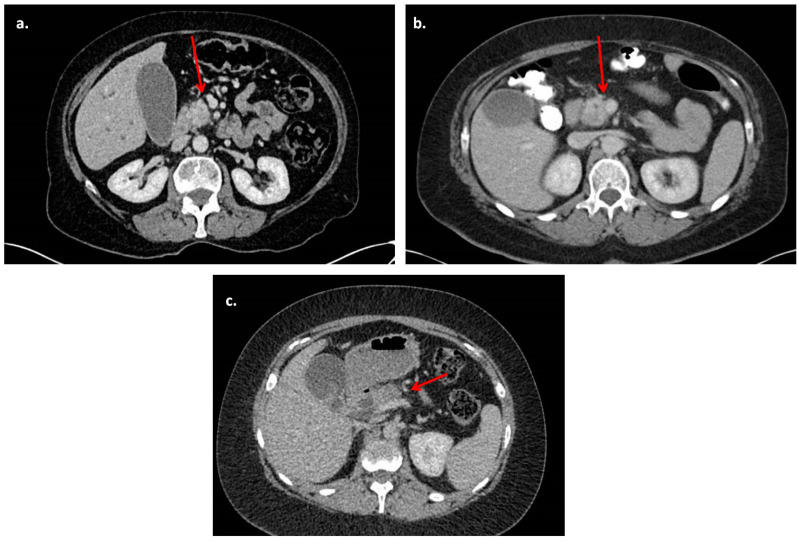
Representative CT scan images. (**a**). Resectable pancreatic uncinate process mass with no venous or arterial abutment (arrows). (**b**). Resectable pancreatic uncinate process mass with abutment of the superior mesenteric vein of <180 degrees and no change in vein wall contour. (**c**). Borderline resectable pancreatic head mass with compression of the portal vein (case not included in this study).

**Table 1 diagnostics-14-00135-t001:** Cohort characteristics.

	Entire Cohort	No Vein Resection (n = 234)	Vein Resection (n = 62)	*p*
Age (median; IQR)	69 (63–74)	69 (63–74)	73 (67–77)	**0.019**
Gender (male:female)	154:142 (52%:48%)	129:105 (55.1%:44.9%)	25:37 (40.3%:59.7%)	**0.045**
BMI (median; IQR)	25.9 (22.8–29.3)	25.5 (22.9–29.7)	26 (22.5–29)	0.862
Pathology				
pT				0.618
pT1	46 (15.5%)	35 (15%)	11 (17.7%)	
pT2	230 (77.7%)	183 (78.2%)	47 (75.8%)	
pT3	20 (6.8%)	16 (6.8%)	4 (6.5%)	
pN				0.251
pN0	38 (12.8%)	32 (13.7%)	6 (9.7%)	
pN1	109 (36.8%)	88 (37.6%)	21 (33.9%)	
pN2	149 (50.3%)	114 (48.7%)	35 (56.4%)	
Lymph node yield (median; IQR)				
Total	19 (14–24)	19 (13–23)	20 (16–24)	0.059
Positive	4 (1–7)	3 (1–7)	4 (2–6)	0.868
Positive margins (R1)	159 (53.7%)	114 (48.7%)	45 (72.6%)	**0.001**
CT parameters				
Tumour size (mm) (median; IQR)	20 (10–27)	21 (12–28)	19 (0–26)	0.629
Tumour location				0.582
Not visible	18 (6.1%)	15 (6.4%)	3 (4.8%)	
Head	196 (66.2%)	151 (64.5%)	45 (72.6%)	
Uncinate	76 (25.7%)	65 (27.8%)	11 (17.8%)	
Neck	6 (2%)	3 (1.3%)	3 (4.8%)	
Pancreatitis	35 (11.8%)	24 (10.3%)	11 (17.7%)	0.122
CBD dilatation	279 (94.3%)	219 (93.6%)	60 (96.8%)	0.389
PD dilatation	218 (73.6%)	168 (71.8%)	50 (80.6%)	0.195
PD size (mm) (median; IQR)	5 (3–8)	5 (3–7)	6 (4–8)	0.147
Lymphadenopathy	106 (35.8%)	85 (36.3%)	21 (33.9%)	0.767
PV involvement	38 (12.8%)	24 (10.3%)	14 (22.6%)	0.013
PV involvement length (mm) (median; IQR)	0 (0–0)	0 (0–0)	0 (0–0)	0.012
PV involvement				
Circumference				
No contact	256 (86.5%)	208 (88.9%)	48 (77.4%)	0.019
0–90 degrees	34 (11.5%)	22 (9.4%)	12 (19.4%)	
91–180 degrees	6 (2%)	4 (1.7%)	2 (3.2%)	
Confluence involvement	50 (16.9%)	35 (15%)	15 (24.2%)	0.089
SMV involvement	93 (31.4%)	70 (29.9%)	23 (37.1%)	0.285
SMV involvement length (mm) (median; IQR)	0 (0–10)	0 (0–9)	0 (0–14)	0.098
SMV involvement				
Circumference				
No contact	206 (69.6%)	167 (71.4%)	39 (62.9%)	0.129
0–90 degrees	78 (26–4%)	61 (26.1%)	17 (27.4%)	
91–180 degrees	12 (4.1%)	6 (2.5%)	6 (9.7%)	
Time CT to operation (days) (median; IQR)	36 (16–72)	35 (16–72)	41 (19–72)	0.795

**Table 2 diagnostics-14-00135-t002:** Time-to-event analysis of CT parameters as predictors for vein resection.

Univariate Analysis		
**CT Parameters**	**Hazard Ratio**	**95% CI**
Tumour size (mm)	0.343	1.009 (0.991–1.027)
Tumour size (categorical)	0.939	
0–20 mm	Indicator	
20–40 mm	0.919	0.973 (0.578–1.639)
>40 mm	0.755	1.181 (0.415–3.360)
Tumour location	0.142	
Not visible	Indicator	
Head	0.139	2.427 (0.750–7.848)
Uncinate	0.308	1.950 (0.541–7.032)
Neck	**0.025**	**6.272 (1.253–31.396)**
Pancreatitis	0.527	1.237 (0.640–2.389)
CBD dilatation	0.122	3.070 (0.742–12.706)
PD dilatation	0.063	1.823 (0.969–3.428)
PD size (mm)	0.085	1.063 (0.992–1.139)
Lymphadenopathy	0.089	1.607 (0.930–2.780)
PV involvement	**0.035**	**1.914 (1.047–3.501)**
PV involvement length (mm)	0.171	1.023 (0.990–1.056)
PV involvement length ≥ 10 mm	**0.044**	**1.856 (1.017–3.385)**
PV involvement circumference		
<90 degrees	Indicator	
90–180 degrees	**0.003**	**3.727 (1.583–8.774)**
Confluence involvement	0.069	1.731 (0.959–3.125)
SMV involvement	0.073	1.607 (0.958–2.696)
SMV involvement length (mm)	0.174	1.015 (0.993–1.037)
SMV involvement length ≥ 10 mm	**0.043**	**1.716 (1.018–2.894)**
SMV involvement circumference		
<90 degrees	Indicator	
90–180 degrees	**0.003**	**3.727 (1.583–8.774)**
**Multivariate analysis**		
**CT Parameters**	**Hazard ratio**	**95% CI**
Tumour location	0.057	
Not visible	Indicator	
Head	0.065	3.049 (0.932–9.976)
Uncinate	0.094	3.085 (0.826–11.513)
Neck	**0.006**	**9.870 (1.921–50.706)**
SMV involvement length ≥ 10 mm	**0.004**	**3.513 (1.477–8.352)**
Lymphadenopathy	0.077	1.652 (0.946–2.883)
CBD dilatation	0.096	3.422 (0.802–14.596)
PD dilatation	**0.045**	**1.929 (1.015–3.667)**

CBD: common bile duct; PD: pancreatic duct, PV: portal vein; SMV: superior mesenteric vein.

**Table 3 diagnostics-14-00135-t003:** Time-to-event analysis of CT parameters as predictors for venous resection with interposition graft reconstruction.

Univariate Analysis		
**CT Parameters**	**Hazard Ratio**	**95% CI**
Tumour size (mm)	0.633	1.013 (0.962–1.066)
Tumour size (categorical)	0.609	
0–20 mm	Indicator	
20–40 mm	0.560	0.602 (0.110–3.311)
>40 mm	0.524	2.052 (0.224–18.769)
Tumour location	0.962	
Not visible	Indicator	
Head	0.969	87,950 (0–3 × 10^252^)
Uncinate	0.970	49,181.527 (0–2 × 10^252^)
Neck	1.000	1.034 (0)
Pancreatitis	0.057	4.296 (0.956–19.315)
CBD dilatation	0.581	23.959 (0–1,877,946)
PD dilatation	0.307	35.500 (0.038–33,452.052)
PD size (mm)	0.744	1.034 (0.844–1.267)
Lymphadenopathy	0.220	2.685 (0.555–12.999)
PV involvement	0.894	1.155 (0.138–9.636)
PV involvement length (mm)	0.437	1.033 (0.951–1.123)
PV involvement length ≥ 10 mm	0.932	1.097 (0.132–9.115)
PV involvement circumference		
<90 degrees	Indicator	
90–180 degrees	0.812	0.048 (0–4 × 10^9^)
Confluence involvement	0.340	2.234 (0.428–11.656)
SMV involvement	0.940	1.065 (0.206–5.512)
SMV involvement length (mm)	0.740	1.011 (0.947–1.079)
SMV involvement length ≥ 10 mm	0.803	1.233 (0.239–6.371)
SMV involvement circumference		
<90 degrees	Indicator	
90–180 degrees	0.763	0.047 (0–2 × 10^7^)
**Multivariate analysis**		
**As only one parameter with *p* < 0.200, multivariate model was not performed**		

CBD: common bile duct; PD: pancreatic duct, PV: portal vein; SMV: superior mesenteric vein.

## Data Availability

The data presented in this study are available on request from the corresponding author. The data are not publicly available due to data protection reasons.

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
