# Peer review of "Staging Computed Tomography Parameters Predict the Need for Vein Resection during Pancreaticoduodenectomy in Resectable Pancreatic Ductal Adenocarcinoma"

_diagnostics, 2024, doi:10.3390/diagnostics14020135_

Round 1
Reviewer 1 Report
Comments and Suggestions for Authors
The manuscript is the study on prediction of the need for portal vein resection during pancreaticoduodenectomy using computed tomography in resectable pancreatic ductal adenocarcinoma.
The authors have concluded that staging CT parameters can predict the need for venous resection during PD for resectable cases of PDAC.
The study is interesting.
There are several comments on this manuscript.
1. In Table 2. multivariate analysis showed outcomes of SMV involvement, but PV involvement is not included. Please discuss on these.
2. In Table 1. Cohort characteristics, in the row of PV involvement length (mm), the values are same in 3 columns. Please confirm these.
3. Interposition graft reconstruction was performed in only 7 cases (11%).
Therefore, it is too small numbers to draw any conclusion.
Necessary of Table 3, should be reconsidered.
4. In Reference 8, there is duplication of author name.
Author Response
The authors thank the reviewers for their time and comments. Detailed point by point reply is provided below.
Reviewer 1
- In Table 2. multivariate analysis showed outcomes of SMV involvement, but PV involvement is not included. Please discuss on these.
In table 2 multivariate analysis section only parameters that approached or achieved statistical significance were included. PV involvement did not and hence not included.
- In Table 1. Cohort characteristics, in the row of PV involvement length (mm), the values are same in 3 columns. Please confirm these.
The values used are correct. The median and IQR values were the same for all 3 categories. The words “(median; IQR)” were introduced next to the relevant parameters in the first column to make this more understandable to the readers
- Interposition graft reconstruction was performed in only 7 cases (11%). Therefore, it is too small numbers to draw any conclusion. Necessary of Table 3, should be reconsidered.
The authors agree with the comment. In the discussion section in the relevant paragraph it is mentioned that with larger numbers the presence of pancreatitis would reach significance (“and it is likely that with larger numbers this would have reached significance”).
- In Reference 8, there is duplication of author name.
Corrected
Reviewer 2 Report
Comments and Suggestions for Authors
Author Response
The authors thank the reviewers for their time and comments. Detailed point by point reply is provided below.
Reviewer 2
- The reviewer believes that the author should change the title of the manuscript to better match the conclusions presented.
This is an interesting comment. As there are 3 parameters that have been identified as predictive factors, including them in the title would make it unnecessarily long and detailed. The authors respectfully would keep the existing title.
- Fig. 1 No meaning for readers. The author should zoom in on the area that needs to be presented and describe that area in more detail.
Figure 1 was introduced to the manuscript following the advice of the editorial team. It presents characteristic images from various stages of pancreatic cancer. The specific area of interest is signified by the arrows and it is explained in the figure legend. The jpeg images used have been provided to the editorial team and the use of larger images (in which the areas relating to the arrows are more clearly visible) may be possible depending on the journals publication policies.
- Table 1,2,3 contains too many parameters, and it is difficult to consider all of them. Authors should focus only on parameters directly related to the conclusions in this manuscript. In particular, the author should have a deeper analysis of these parameters.
The paper has followed the common practice in scientific literature, which is for all parameters analysed to be presented in tables and then only the important ones to be analysed in the manuscript. Tables 1-3 include all the statistical analysis of the study. The type of statistical analysis is explained in the relevant section of the manuscript. The parameters that were significant have been presented and analysed in detail in the manuscript.